# Anti-Inflammatory Effects of Miyako *Bidens pilosa* in a Mouse Model of Amyotrophic Lateral Sclerosis and Lipopolysaccharide-Stimulated BV-2 Microglia

**DOI:** 10.3390/ijms241813698

**Published:** 2023-09-05

**Authors:** Komugi Tsuruta, Takato Shidara, Hiroko Miyagishi, Hiroshi Nango, Yoshihiko Nakatani, Naoto Suzuki, Taku Amano, Toyofumi Suzuki, Yasuhiro Kosuge

**Affiliations:** 1Laboratory of Pharmacology, School of Pharmacy, Nihon University, 7-7-1 Narashinodai, Funabashi 274-8555, Chiba, Japan; phko22004@g.nihon-u.ac.jp (K.T.); phta14114@g.nihon-u.ac.jp (T.S.); miyagishi.hiroko@nihon-u.ac.jp (H.M.); phhi16003@g.nihon-u.ac.jp (H.N.); 2Department of Pharmacotherapeutics, School of Pharmacy, International University of Health and Welfare, 2600-1 Kitakanemaru, Ohtawara 324-8501, Tochigi, Japan; ynakatani@iuhw.ac.jp; 3Laboratory of Pharmaceutics, School of Pharmacy, Nihon University, 7-7-1 Narashinodai, Funabashi 274-8555, Chiba, Japan; suzuki.naoto65@nihon-u.ac.jp (N.S.); suzuki.toyofumi@nihon-u.ac.jp (T.S.); 4Tochigi Prefectural Okamotodai Hospital, 2162 Shimookamotomachi, Utsunomiya 329-1104, Tochigi, Japan; tamano01@okamotodai.jp

**Keywords:** amyotrophic lateral sclerosis, *Bidens pilosa*, pro-inflammatory cytokines, spinal cord, M1-microglia/macrophages

## Abstract

Neuroinflammation is a fundamental feature in the pathogenesis of amyotrophic lateral sclerosis (ALS) and arises from the activation of astrocytes and microglial cells. Previously, we reported that Miyako *Bidens pilosa* extract (MBP) inhibited microglial activation and prolonged the life span in a human ALS-linked mutant *superoxide dismutase-1* (*SOD1*^G93A^) transgenic mouse model of ALS (G93A mice). Herein, we evaluated the effect of MBP on microglial activation in the spinal cord of G93A mice and lipopolysaccharide-stimulated BV-2 microglial cells. The administration of MBP inhibited the upregulation of the M1-microglia/macrophage marker (interferon-γ receptor (IFN-γR)) and pro-inflammatory cytokines (tumor necrosis factor (TNF)-α, interleukin (IL)-1β, and IL-6) in G93A mice. However, MBP did not affect the increase in the M2-microglia/macrophage marker (IL-13R) and anti-inflammatory cytokines (transforming growth factor (TGF)-β and IL-10) in G93A mice. BV-2 cell exposure to MBP resulted in a decrease in 3-(4,5-dimethylthiazol-2-yl)-2,5-diphenyl tetrazolium (MTT) reduction activity and bromodeoxyuridine incorporation, without an increase in the number of ethidium homodimer-1-stained dead cells. Moreover, MBP suppressed the production of lipopolysaccharide-induced pro-inflammatory cytokines (TNF-α, IL-1β, and IL-6) in BV-2 cells. These results suggest that the selective suppression of M1-related pro-inflammatory cytokines is involved in the therapeutic potential of MBP in ALS model mice.

## 1. Introduction

Amyotrophic lateral sclerosis (ALS) is a fast-progressing and devastating neurodegenerative disease characterized by motor neuron death in the spinal cord, brain stem, and motor cortex. Most cases of ALS are of unknown etiology, classified as sporadic ALS, while approximately 5–10% of the cases are classified as familial ALS. Although more than 20 ALS genes have been identified, mutations of the *copper-zinc (Cu/Zn) superoxide dismutase* (*SOD1*) gene were the first shown to cause familial ALS [1]. In ALS research, transgenic mice overexpressing high levels of a mutant human SOD1 containing a glycine-to-alanine substitution at position 93 (*SOD1*^G93A^) (G93A mice) are used as the standard model [2]. Several studies using G93A mice have revealed some mechanisms of pathogenesis in motor neuron death such as oxidative stress [3], protein aggregation [4], and neuroinflammation [5]. Recently, attention has been focused on the direct contribution of non-neuronal cells, such as astrocytes and microglia, to motor neuron death in ALS [6]. Indeed, the selective reduction in mutant *SOD1* from microglia [7] or astrocytes [8] in mice using the Cre–Lox system slowed disease progression in *SOD1*^G37R^ ALS model mice. Thus, the importance of microglia and astrocytes as non-cell autonomous players is widely recognized in the pathological study of ALS.

Microglia are the resident macrophages in the central nervous system (CNS) and are gaining recognition for their critical role in CNS immune surveillance. Microglia-associated neuroinflammation is a pathological feature in both patients with ALS and mouse models of the disease [9], and positron emission tomography studies have shown that activated microglia are existent in the brains of living patients with ALS [10,11]. Autopsies of patients with ALS revealed that activated microglia exist around the injury sites of motor neurons in spinal cord tissue [12]. Furthermore, extracellular mutant SOD1 from G93A mice does not cause direct disability in cultured motor neurons, but it activates microglia, which then enhance motor neuron death through producing neurotoxic factors including proinflammatory molecules and free radicals [13,14]. Previous studies showed that treatment with minocycline or *Hirsutella sinensis* reduced the level of activated microglia and the degeneration of motor neurons and also prolonged the lifespan in G93A mice, suggesting that microglial regulation may represent an effective treatment for ALS [15,16,17]. Interestingly, replacing wild-type microglia/macrophage via bone marrow transplantation slowed disease progression in G93A mice [18]. These results suggest that microglia/macrophages may be a more useful target than astrocytes in the regulation of ALS neuroinflammation. Increasing evidence shows that microglia/macrophages in the CNS are categorized by two opposite types, the M1 and M2 phenotypes. Generally, M1 microglia/macrophages induce inflammation and neurotoxicity through releasing inflammatory mediators, while M2 microglia/macrophage induce anti-inflammation and neuroprotectivity via the production of anti-inflammatory mediators [19,20]. The microglia/macrophage phenotype changes depending on the disease severity and stage [19,20]. Therefore, a therapeutic approach to modulate microglia/macrophage phenotypes, either enhancing anti-inflammatory M2 phenotypic properties or reducing the M1 phenotype depending on toxicity, could be a potential treatment strategy for ALS.

*Bidens pilosa* L. var. *radiata* SCHERFF (BP) is an annual herb, belonging to the Asteraceae family that originated from South America and can now be found in tropical and subtropical regions around the world [21]. Miyako BP extract (MBP) is cultivated with only green manure in the Miyako Islands of Okinawa Prefecture, Japan, and is used as an herbal tea ingredient and in various folk medicines. MBP has been reported to have beneficial effects including anti-inflammatory [22], antioxidant [23], and antiallergy properties [22]. More recently, we reported that MBP is capable of prolonging the life span of G93A mice when administered at an oral gavage of 2 g/kg/day from the post-symptomatic stage to the end stage of the disease [24]. Importantly, MBP attenuated the activation or induction of microglia in the spinal cord of G93A mice [24]. In this study, we examined whether MBP could regulate microglial polarization in the spinal cord of G93A mice and lipopolysaccharide (LPS)-stimulated BV-2 microglial cells.

## 2. Results

### 2.1. MBP Selectively Reduced the Production of M1-Related Pro-Inflammatory Cytokines in the Lumbar Spinal Cord of G93A Mice

M1-phenotype microglia/macrophage represent pro-inflammatory activity and secrete various pro-inflammatory cytokines such as tumor necrosis factor-α (TNF-α), interleukin-1β (IL-1β), and interleukin-6 (IL-6), whereas M2-phenotype microglia/macrophage promote anti-inflammation and tissue repair and produce anti-inflammatory cytokines such as transforming growth factor-β (TGF-β) and interleukin-10 (IL-10) [19]. We performed real-time PCR to evaluate the mRNA level of pro-inflammatory and anti-inflammatory cytokines in the lumbar spinal cord of G93A mice. Our real-time PCR results indicated that the mRNA levels of M1-related pro-inflammatory cytokines (TNF-α, IL-1β, and IL-6) were significantly increased in G93A mice compared with that of vehicle-treated WT mice (Figure 1a). The upregulation of inflammatory cytokines in G93A mice was partially but significantly reversed via the oral administration of MBP (Figure 1a). Additionally, the mRNA level of M2-associated-anti-inflammatory cytokines (TGF-β and IL-10) was increased in the spinal cord of vehicle-treated G93A mice compared with that of vehicle-treated WT mice (Figure 1b). Unlike M1-related pro-inflammatory cytokines, the treatment of MBP did not inhibit an increase in the supply of anti-inflammatory cytokines in G93A mice (Figure 1b).

### 2.2. MBP Selectively Inhibited M1-Microglia/Macrophage Polarization in the Lumbar Spinal Cord of G93A Mice

The activated phenotype of microglia/macrophages is classified by the production pattern of cytokines: the pro-inflammatory-related M1-phenotype and the anti-inflammatory-activated M2-phenotype microglia/macrophages [19]. To investigate whether MBP affected the phenotype of microglia/macrophages, real-time PCR was performed and the expression level of the main polarization markers was analyzed for M1 (interferon-γ receptor; IFN-γR) and M2 (interleukin-13 receptor; IL-13R) microglia/macrophages in the lumbar spinal cord of G93A and WT mice. The mRNA level of the M1-phenotype microglia/macrophage marker IFN-γR was significantly increased in the vehicle-treated G93A mice compared with that of the vehicle-treated WT mice, while it was clearly reduced in the MBP-treated G93A mice (Figure 2a). Although the mRNA level of the M2-phenotype microglia/macrophage marker IL-13R in the vehicle-treated G93A mice was also higher than that in the vehicle-treated WT mice, the upregulation of IL-13R in G93A mice did not change after the treatment of MBP (Figure 2b). Moreover, MBP had no marked effect on both M1- and M2-phenotypic microglia/macrophage markers in WT mice (Figure 2).

### 2.3. MBP Suppressed the Cell Proliferation of BV-2 Microglial Cells

We further examined the effect of exogenously applied MBP on the cell proliferation of microglia using the mouse microglial cell line BV-2. As shown in Figure 3, the exposure of BV-2 cells to 2–1000 µg/mL of MBP for 24 h resulted in a concentration-dependent decrease in 3-(4,5-dimethylthiazol-2-yl)-2,5-diphenyl tetrazolium (MTT) reduction activity (Figure 3b). A statistically significant decrease in MTT reduction activity was observed at a concentration of 20 µg/mL (80.1 ± 2.2%, *p* < 0.001), 100 µg/mL (64.8 ± 4.5%, *p* < 0.0001), 200 µg/mL (48.9 ± 3.1%, *p* < 0.0001), and 1000 µg/mL (33.8 ± 2.3%, *p* < 0.0001) compared to that of vehicle-treated cells (101.4 ± 1.5%). On the other hand, the bromodeoxyuridine (BrdU) incorporation assay showed that only the bath application of 1000 µg/mL MBP caused a significant decrease in BrdU incorporation into BV-2 cells compared with that of vehicle-treated cells (Figure 3c). Moreover, observation via phase contrast microscopy demonstrated that treatment with MBP for 24 h at any tested concentration did not lead to an increase in the cell displaying cell death-related morphological characteristics including blebbing, cell shrinkage, and detachment of the adherent cells from the plate (Figure 3d).

### 2.4. MBP Did Not Induce Cell Death in BV-2 Cells

Next, we investigated cell mortality after treatment with MBP in BV-2 cells with the Live/Dead cell staining kit using the time schedule as shown in Figure 4a. The exposure of BV-2 cells to 10–1000 μg/mL MBP for 24 h did not change the percentage of dead cells stained by EthD-1. The percentage of EthD-1-positive cells in the 10, 100, and 1000 μg/mL MBP-treated cells was 2.7 ± 0.5%, 3.1 ± 0.8%, and 8.6 ± 2.6%, respectively. The total cell number of 1000 μg/mL MBP-treated cells was significantly reduced to approximately 75% of that in the vehicle-treated cells (Figure 4b). Taken together, these results suggest that the decrease in MTT reduction activity induced by the high concentration of MBP is not involved in the cytotoxicity of MBP on BV-2 cells.

### 2.5. MBP Inhibited LPS-Induced Pro-Inflammatory Cytokine Production in BV-2 Cells

In order to investigate the potential role of MBP in the LPS-induced production of pro-inflammatory cytokines (TNF-α, IL-1β, and IL-6), BV-2 cells were treated with 100 and 1000 μg/mL MBP with or without 1 μg/mL LPS for 4 h. Real-time PCR analysis demonstrated that the level of pro-inflammatory cytokines was dramatically upregulated by LPS stimulation, as evidenced by the increase in the mRNA levels of TNF-α, IL-1β, and IL-6 compared to that of untreated cells (Figure 5). As shown in Figure 5, MBP treatment significantly suppressed the LPS-induced production of pro-inflammatory cytokines. Additionally, although no statistical significance was found, the mRNA levels of pro-inflammatory cytokines pretended to increase in the MBP-alone-treated BV-2 cells (Figure 5).

## 3. Discussion

ALS is a progressive and lethal neurodegenerative disease of the motor neurons with no effective treatment or cure to date. Therefore, there is a need to create novel disease-modifying strategies to suppress disease progression and extend the lifespan of patients with ALS. Previously, we reported that the oral administration of MBP after the onset of ALS-like symptoms extended survival duration, improved motor performance, and protected motor neurons through the inhibition of astrocyte and microglia activation in G93A mice [24]. In this study, we demonstrated for the first time that MBP selectively reduced the production of pro-inflammatory cytokines through inhibiting the polarization of M1 microglia/macrophages in the lumbar spinal cord of G93A mice. We also observed that exposure of BV-2 cells to MBP suppressed LPS-induced pro-inflammatory cytokine production and cell proliferation without affecting the number of EthD-1-positive dead cells. Overall, the results show that MBP restored microglia/macrophage balance not only through inhibiting the polarization of microglia/macrophages to the M1-like phenotype but also through regulating cell proliferation.

Neuroinflammation caused by pro-inflammatory cytokines is one of the most striking pathological hallmarks of ALS [25]. In the lumbar spinal cord of G93A mice, microglial activation was detected at 15 weeks (105 days) of age and older [26], which also correlated with transcriptional changes of genes involved in inflammation [27]. M1-phenotype microglia/macrophage produce a violent inflammatory response characterized by the production of pro-inflammatory cytokines (TNF-α, IL-1β, and IL-6) and increased expression of IFN-γR [16,28,29,30]. On the other hand, the M2 microglia/macrophage phenotype leads to the release of anti-inflammatory cytokines (TGF-β and IL-10) and increased expression of IL-13R [16,28,31]. Our present study found that both M1- and M2-phenotype markers were increased in the spinal cord of G93A mice at an early symptomatic stage (16 weeks of age). IFN-γR and IL-13 receptors are not only expressed on microglia, but also on B cells [32] and a number of immune cells [33]. In periods of neuroinflammation, the blood–spinal barrier becomes permeable, facilitating the entry of peripheral macrophages, monocytes, T cells, and B cells into the spinal cord of both ALS mice and patients [34,35]. In these cases, microglia and macrophages are indistinguishable; therefore, we use the collective term “microglia/macrophage” in this study. It has been reported that the suppression of M1-polarized microglia might be beneficial to the protection of motor neurons and prevention of disease progression in G93A mice. Indeed, treatment with minocycline before the motor dysfunction of G93A mice significantly prolonged survival via the selective inhibition of M1 microglia polarization without affecting the neuroprotective M2 microglia [16]. Moreover, treatment with *Hirsutella sinensis* suppressed M1 microglial polarization and motor neuron death in the spinal cord of G93A mice [17]. Our results clearly demonstrated that MBP selectively downregulated M1 microglia/macrophage polarization without affecting the increased state of M2 microglia/macrophage polarization in G93A mice even after post-onset administration (Figure 2). Furthermore, it has been reported that M2 microglia/macrophages from transplanted donor marrow delayed onset motor dysfunction in G93A mice [36]. These results suggest that the ability of MBP to not only maintain the neuroprotective M2 microglia/macrophage polarization but also to prevent the neurotoxic M1 microglia/macrophage polarization could be involved in improving the motor neuron loss and disease progression of G93A mice.

It has been reported that the levels of pro-inflammatory cytokines such as IFN-γ, TNF-α, IL-1β, C-C motif chemokine ligand-2 (also known as monocyte chemotactic protein-1), C-C motif chemokine ligand-5 (also called RANTES), C-X-C motif chemokine ligand 10 (also known as interferon γ-induced protein 10), and IL-17A increased with aging and disease progression in G93A mice, whereas anti-inflammatory cytokines including IL-4, IL-10, and TGF-β reached peak levels at an early symptomatic stage (90 days of age) and rapidly decreased until 120 days of age (at the symptomatic stage) [37]. On the other hand, previous studies have shown that the increased expression of pro-inflammatory cytokines such as TNF-α, IL-1β, and IL-6 [16,17,38,39] was accompanied by the increased expression of anti-inflammatory cytokines including TGF-β and IL-10 [16,37] at the symptomatic stage in G93A mice. Consistent with previously reported results, we demonstrated that the mRNA levels of pro-inflammatory cytokines (TNF-α, IL-1β, and IL-6) and anti-inflammatory cytokines (TGF-β and IL-10) were increased in the spinal cord of G93A mice at an early symptomatic stage (Figure 1). Erythropoietin dramatically decreased the level of pro-inflammatory cytokines and highly maintained the upregulation of anti-inflammatory cytokines in the spinal cord of G93A mice until the symptomatic stage [37]. Recombinant human erythropoietin has also been reported to prolong the life span and protect the motor neurons of G93A mice [40]. Moreover, rofecoxib, an anti-inflammatory drug, has been reported to decrease the expression of TNF-α and IL-1β in the spinal cord and to prolong survival in G93A mice [41], suggesting that it may be important to reduce pro-inflammatory cytokines in ALS therapy. Consistent with the effects of erythropoietin and rofecoxib, we found that MBP selectively suppressed the upregulation of pro-inflammatory cytokines with no influence on anti-inflammatory cytokine expression in the spinal cord of G93A mice at an early symptomatic stage (Figure 1). Our findings demonstrate that LPS induced the overproduction of pro-inflammatory cytokines (TNF-α, IL-1β, and IL-6) in BV-2 cells (Figure 5). This suggests that LPS-treated BV-2 cells are a cytotoxic M1-type microglia model rather than an M2-type model. In the present study, MBP inhibited LPS-induced pro-inflammatory cytokine gene expression in BV-2 cells (Figure 5). *Hirsutella sinensis* selectively suppressed M1 microglial polarization and M1-related pro-inflammatory cytokine expression (TNF-α, IL-6, and IL-1β) in the spinal cord of G93A mice [17]. Minocycline also possessed a selective inhibitory effect on the increase in M1-polarized microglia as well as the production of pro-inflammatory cytokines (IL-1β, TNF-α, and IFN-γ) in G93A mice [16]. Taken together, these results suggest that the suppression of pro-inflammatory cytokines in G93A mice may be due to the inhibitory effect of MBP on M1 microglial polarization. Additionally, we found that MBP tended to possess weak pro-inflammatory effects in BV-2 cells under basal conditions without LPS treatment (Figure 5), but not in the vehicle-treated WT mice (Figure 1). Future studies are required to determine whether the MBP-induced activation of pro-inflammatory cytokines contributes to the therapeutic mechanism of MBP for ALS.

Generally, an increase in microglial proliferation has been shown to be associated with inflammatory processes in ALS [39]. We demonstrated that the exposure of BV-2 cells to MBP decreased the MTT reduction activity and uptake of BrdU incorporation, suggesting that MBP had an inhibitory effect on cell proliferation activity with no impact on the cell death of BV-2 cells (Figure 3 and Figure 4). These findings suggest that the inflammation-regulating effect of MBP on pro-inflammatory cytokines might be involved in not only the inhibition of M1 microglia/macrophage polarization but also the suppression of microglial proliferation. Moreover, it has been reported that cell cycle inhibitors, including roscovitine and flavopiridol, inhibited microglial proliferation and protected the neuron in animal models of neuroinflammation-related diseases, such as in a rat model of spinal cord injury [42], a mouse model of multiple sclerosis [43], and a rat model of status epilepticus [44]. Because BV-2 microglial cell lines differ from primary microglia in morphology and activation state, additional studies are required to confirm the specificity of MBP’s effect on microglia and macrophages, as well as the mechanism regulating this effect, using more biologically relevant models. The results from these previous models and our mouse model of ALS suggest that the suppression of microglial proliferation might be involved in the desirable protective potential of MBP on motor neurons in G93A mice.

One of the limitations of our present study is that we could not identify any active compound associated with the anti-inflammatory effect of MBP. MBP has been reported to contain caffeic acid, some chlorogenic acids such as neochlorogenic acid and chlorogenic acid, and various flavonoids including rutin, quercetin, quercetin derivatives, hyperin, isoquercitrin, centaurein, and jacein [45,46]. We speculate that the various active compounds in MBP interact with each other to regulate microglia/macrophage-mediated neuroinflammation in the spinal cord of G93A mice. Although further studies will be necessary to identify the specific active compounds present in MBP, the development of pharmacological agents targeting inflammation associated with M1-phenotype microglia/macrophages using ethnomedicine and functional foods could be a potential strategy for treating ALS.

In conclusion, the therapeutic effects of MBP against ALS may be mediated via anti-inflammation through the suppression of microglia/macrophage polarization into M1 microglia/macrophages (Figure 6). More studies are required to identify the molecular targets of these effects by MBP. The findings of this study indicate the promising therapeutic potential of MBP for neurodegenerative diseases involving activated microglia, including ALS.

## 4. Materials and Methods

### 4.1. Animals

All experimental protocols were performed according to the guidelines for animal experiments at Nihon University, accredited by the Ministry of Education, Culture, Sports, Science, and Technology, Japan. Every attempt was made to minimize animal suffering, reduce the number of animals used, and utilize alternatives to in vivo techniques. Transgenic G93A mice were purchased from Jackson Laboratory (Bar Harbor, ME, USA) and maintained as hemizygotes through mating transgenic males with non-transgenic females. The offspring were genotyped using a polymerase chain reaction (PCR) with DNA from venous blood samples. The non-transgenic littermates were used as the control group (WT mice). We used a total of 20 mice divided into the following four groups: vehicle-treated WT mice (*n* = 6), MBP-treated WT mice (*n* = 5), vehicle-treated G93A mice (*n* = 4), and MBP-treated G93A mice (*n* = 5). G93A mice show disease onset at approximately 15 weeks of age and die approximately 4–5 weeks post-onset [26]. These mice were housed under standard conditions (temperature 22 °C, 12 h light/dark cycle) with free access to food and water in the animal center at the School of Pharmacy, Nihon University. Animal experiments were carried out after obtaining approval from the Nihon University Animal Care and Use Committee (AP19PHA026-2 and 2004YAKU003-4).

### 4.2. Cell Culture

The immortalized murine microglial cell line, BV-2, was provided by Professor Eui-Ju Choi (Department of Life Science, Korea University, Seoul, Republic of Korea). Briefly, BV-2 cells were cultured in Dulbecco’s Modified Eagle Medium (DMEM; Life Technologies Corporation, Carlsbad, CA, USA) containing 1% penicillin–streptomycin solution (100×) (Life Technologies Corporation) and 10% fetal bovine serum (Life Technologies Corporation), as described previously [47]. BV-2 cells were seeded at a density of 1.2 × 10^4^ cells/cm^2^ in a culture dish or plate (Asahi Glass Co., Ltd., Shizuoka, Japan). The cells were routinely passaged every three days.

### 4.3. MBP Treatment

MBP (Musashino Miyako BP^®^) was provided free of charge from the Musashino Research Institute for Immunity (Miyako Island, Okinawa, Japan). MBP was dissolved in injection water (Otsuka, Tokyo, Japan). MBP (2 g/kg/day) or a vehicle was administered orally once per day for 1 week at 15 weeks of age in WT and G93A mice, as described previously [24].

### 4.4. Real-Time PCR

Real-time PCR was performed as described previously [48]. BV-2 cells were treated with MBP and then immediately incubated for 4 h with or without 1 μg/mL of LPS (Sigma-Aldrich, St. Louis, MO, USA). Briefly, total RNA was extracted from BV-2 cells and mice lumbar spinal cords using the High Pure RNA Isolation Kit (Roche, Basel, Switzerland). cDNA was synthesized from 0.5 µg of total RNA using reverse transcriptase (Roche). Real-time PCR was performed using TB Green^®^ Premix Ex Taq™ II (TaKaRa, Shiga, Japan). Relative mRNA expression levels were determined using the ΔΔCT method, normalized to glyceraldehyde-3-phosphate dehydrogenase (GAPDH). The primers used for real-time PCR are indicated in Table 1.

### 4.5. MTT Reduction Assay and BrdU Incorporation Assay

The proliferation of BV-2 cells was determined by means of a MTT reduction assay and BrdU incorporation assay. BV-2 cells were treated with MBP in the concentration range of 2–1000 µg/mL for 24 h.

For the MTT assay, BV-2 cells were incubated with MTT reagent (0.25 mg/mL) for 3 h at 37 °C, and the reaction was quenched using a solution containing 50% dimethyl formamide and 20% sodium dodecyl sulfate (pH 4.7) as described previously [49]. The next day, the amount of MTT formazan product in the cells was measured at 570 nm (reference wavelength: 655 nm) with a microplate reader (SH-1000Lab; Corona Electric, Ibaraki, Japan).

The BrdU incorporation assay was performed using the Cell Proliferation ELISA, BrdU Kit (Roche), according to the instruction manual. After labelling the BV-2 cells with BrdU for 24 h, the cells were fixed, and then anti-BrdU-peroxidase was added to the wells to combine with the incorporated BrdU. Following a wash to remove unbound antibody, chromogenic substrate and tetramethyl benzidine was added. The stop solution, 1 M H_2_SO_4_, was added to each well, and the absorbance at 450 nm (reference wavelength: 690 nm) was measured using a microplate reader (SH-1000Lab; Corona Electric).

### 4.6. LIVE/DEAD Assay

The amount of viable and dead cells was evaluated with a LIVE/DEAD^®^ Viability/Cytotoxicity Kit for mammalian cells (Thermo Fisher Scientific, San Diego, CA, USA) using 8-well glass plates (Thermo Fisher Scientific) as described previously [50]. Twenty-four hours after MBP treatment, the BV-2 cells were stained with calcein AM (2 μM) and ethidium homodimer-1 (EthD-1; 4 μM) for 40 min. Fluorescence images were collected with a confocal laser microscope (LSM-710; Zeiss, Oberkochen, Germany). The numbers of calcein-positive live cells and EthD-1-positive dead cells were counted to calculate cell mortality. The cell mortality value indicates the percentage of EthD-1-positive cells to the sum of calcein-positive live cells and EthD-1-positive dead cells (the total number of cells) as described previously [50].

### 4.7. Statistics

Data analyses were evaluated using one-way repeated measure ANOVA followed by Tukey’s test using GraphPad PRISM 9 software (GraphPad Software Inc., San Diego, CA, USA). Results are expressed as the mean ± standard error (SE) or standard deviation (SD).

## Figures and Tables

**Figure 1 ijms-24-13698-f001:**
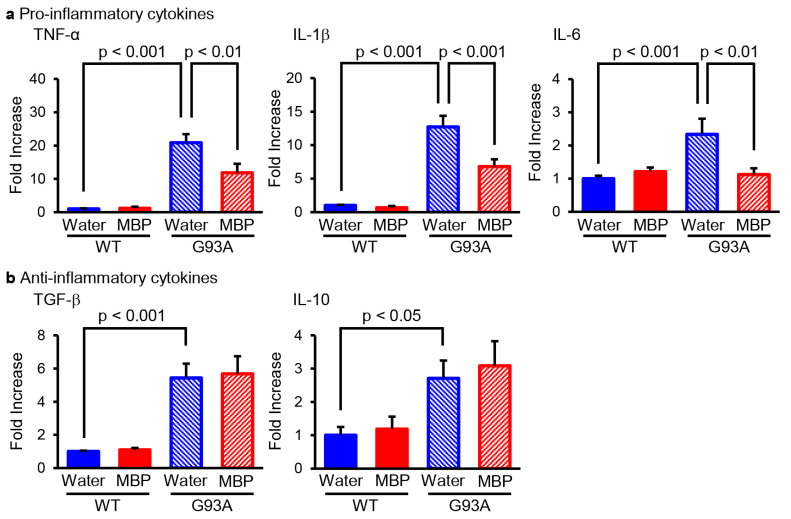
Miyako *Bidens pilosa* (MBP) selectively inhibited the induction of pro-inflammatory cytokines in the lumber spinal cord of G93A mice. Mice were orally administered injection water (vehicle) or MBP (2 g/kg/day), starting at 15 weeks old (asymptomatic stage). One week after the start of treatment, the lumbar spinal cords were analyzed via real-time polymerase chain reaction (PCR). The graphs show the mRNA expression profiles of pro-inflammatory cytokines (tumor necrosis factor (TNF)-α, interleukin (IL)-1β, and IL-6 (**a**)) and anti-inflammatory cytokines (TGF-β and IL-10 (**b**)). Data are shown as the average ± standard error, calculated from 4–6 independent biological replicates, each with 2 technical replicates.

**Figure 2 ijms-24-13698-f002:**
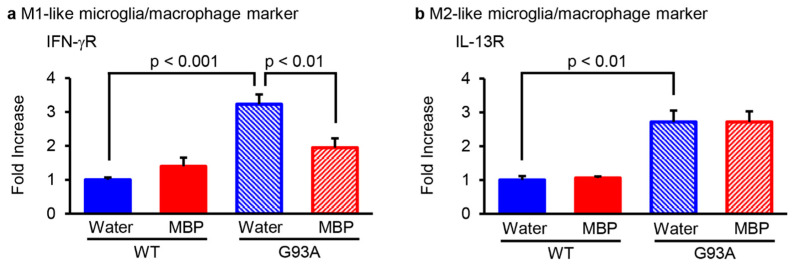
Miyako *Bidens pilosa* (MBP) regulated microglia/macrophage M1–M2 polarization in the lumber spinal cord of WT and G93A mice. Mice were orally administered injection water (vehicle) or MBP (2 g/kg/day), starting at 15 weeks old (asymptomatic stage). One week after the start of treatment, the lumbar spinal cords were analyzed via real-time polymerase chain reaction (PCR). The graphs show the mRNA expression profiles of the M1-like microglia/macrophage marker (interferon (IFN)-γR (**a**) and M2-like microglia/macrophage marker (interleukin (IL)-13R (**b**)) in the spinal cord. Data are shown as the average ± standard error, calculated from 4–6 independent biological replicates, each with 2 technical replicates.

**Figure 3 ijms-24-13698-f003:**
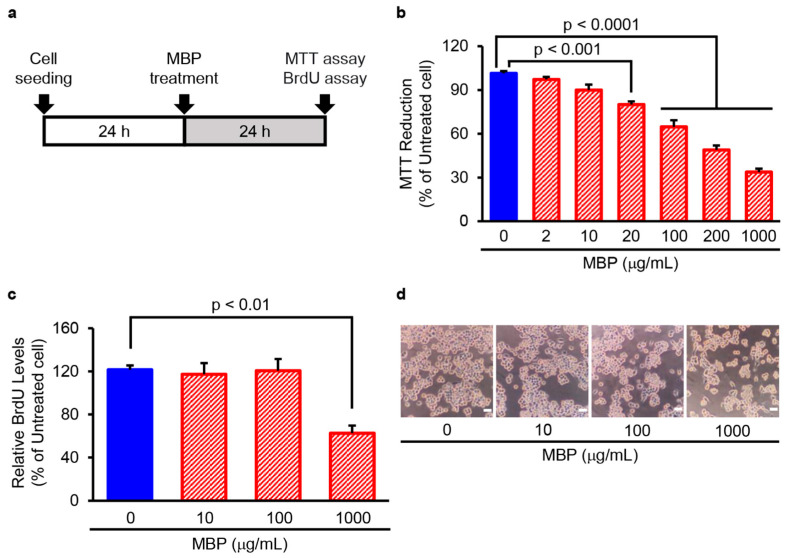
Miyako *Bidens pilosa* (MBP) suppressed the proliferation of BV-2 microglia cells. (**a**) The research protocol conducted in this study is shown. BV-2 cells were treated with the indicated concentrations of MBP for 24 h, and cell proliferation was determined via 3-(4,5-dimethylthiazol-2-yl)-2,5-diphenyl tetrazolium (MTT) reduction assay and bromodeoxyuridine (BrdU) incorporation assay. (**b**) The graph shows the effect of MBP on MTT reduction activity in the cells. Values are indicated as percentages relative to untreated cells. Data are shown as the average ± standard error (SE), calculated from four independent biological replicates, each with four technical replicates. (**c**) The graph shows the effect of MBP on BrdU incorporation into newly synthesized DNA of proliferating cells. Values are indicated as percentages relative to untreated cells. Data are shown as the average ± SE, calculated from four independent biological replicates, each with four technical replicates. (**d**) Cell morphology was observed with an optical microscope; photographs show typical phase-contrast microscopy images of BV-2 cells treated with indicated concentrations of MBP. The scale bar represents 100 μm.

**Figure 4 ijms-24-13698-f004:**
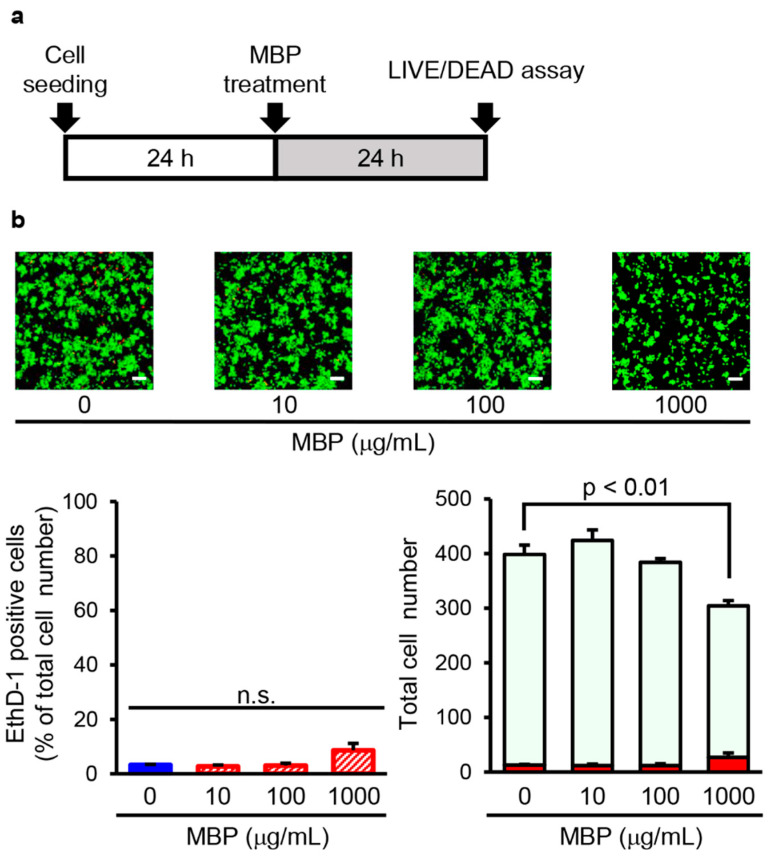
Miyako *Bidens pilosa* (MBP) did not induce cell death in BV-2 cells. (**a**) The research protocol conducted in this study is shown. BV-2 cells were treated with indicated concentrations of MBP for 24 h. (**b**) The photographs show typical fluorescence images of calcein-AM (green, live cells) and ethidium homodimer-1 (EthD-1) (red, dead cells) double staining in each treatment group. The scale bar indicates 100 μm. The left graph shows the percentage of EthD-1-positive dead cells in these cells. The right graph shows the number of EthD-1-positive dead cells (red bar) and calcein-AM-positive live cells (pale green bar) in these cells. n.s.: no significance. Data are shown as the average ± standard error, calculated from four independent biological replicates, each with eight technical replicates.

**Figure 5 ijms-24-13698-f005:**
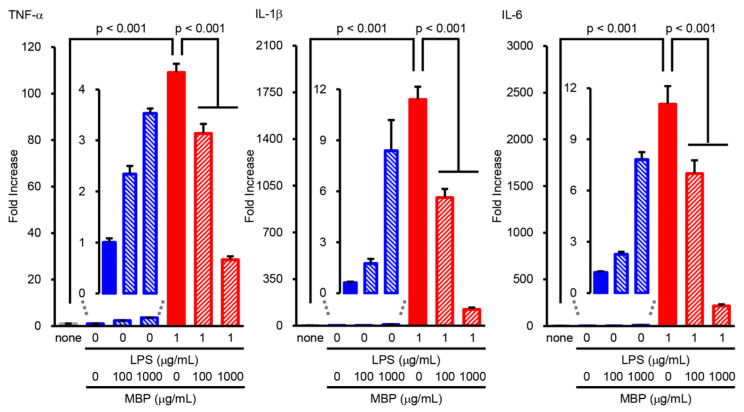
Miyako *Bidens pilosa* (MBP) inhibited the lipopolysaccharide (LPS)-induced production of pro-inflammatory cytokines in BV-2 cells. Cells incubated at different concentrations of MBP with or without LPS (1 μg/mL) for 4 h. The expression level of pro-inflammatory cytokines (tumor necrosis factor (TNF)-α, interleukin (IL)-1β, and IL-6) was analyzed via real-time polymerase chain reaction (PCR). The graphs show the relative expression of mRNA of LPS-induced pro-inflammatory cytokines (TNF-α, IL-1β, and IL-6). Data are shown as the average ± standard error, calculated from six independent biological replicates, each with two technical replicates.

**Figure 6 ijms-24-13698-f006:**
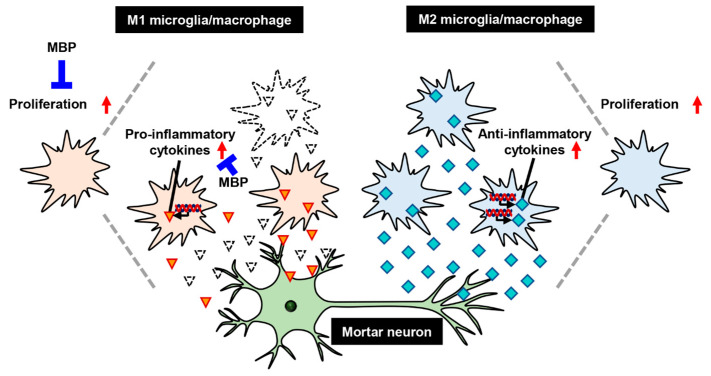
A proposed mechanism of treatment with Miyako *Bidens pilosa* (MBP) in the spinal cord of G93A mice. Red arrows indicate increase of cells proliferation or upregulation of cytokines. Red triangles and sky blue squares indicate pro-inflammatory cytokines and anti-inflammatory cytokines respectively.

**Table 1 ijms-24-13698-t001:** Primer sequences.

Primer	Sequence (5′ to 3′)	
IFN-γR	Forward	TGACGGGAGCACCTGTTACAC
	Reverse	TTTCGACCGTATGTTTCGTATGTAG
IL-13R	Forward	CAGTCTTGCAGCATGGGAACA
	Reverse	TGAGTCCCTAAGGCCTGGAGATTAC
TNF-α	Forward	AGCCCACGTCGTAGCAAACCAC
	Reverse	AGGTACAACCCATCGGCTGGCA
IL-1β	Forward	GCAACTGTTCCTGAACTCAACT
	Reverse	ATCTTTTGGGGTCCGTCAACT
IL-6	Forward	GAGGATACCACTCCCAACAGACC
	Reverse	AAGTGCATCATCGTTGTTCATACA
TGF-β	Forward	TGACGTCACTGGAGTTGTACGG
	Reverse	GGTTCATGTCATGGATGGTGC
IL-10	Forward	GCATGGCCCAGAAATCAAGG
	Reverse	GAGAAATCGATGACAGCGCC
GAPDH	Forward	TGTGTCCGTCGTGGATCTGA
	Reverse	TTGCTGTTGAAGTCGCAGGAG

## Data Availability

The data used to support the findings of this study are available from the corresponding author upon request.

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
