# Peer review of "Anti-Inflammatory Effects of Miyako Bidens pilosa in a Mouse Model of Amyotrophic Lateral Sclerosis and Lipopolysaccharide-Stimulated BV-2 Microglia"

_ijms, 2023, doi:10.3390/ijms241813698_

Round 1
Reviewer 1 Report
The authors of the manuscript titled 'Anti-inflammatory effects of Miyako Bidens pilosa in a mouse model of amyotrophic lateral sclerosis and lipopolysaccharide-stimulated BV-2 microglia' have presented significant findings evaluating the effect of MBP application on microglial activation while identifying its role in selectively suppressing M1-related proinflammatory cytokines. The authors attempt to highlight the therapeutic potential of MBP for prolonging and potentially alleviating the symptoms of ALS. Overall, the results are systematically presented and the experiments are easy to follow and interpret. The writing is systematic and thoroughly referenced to validate the current observations in light of previous research in the field. The concerns raised while reviewing the manuscript have been listed below:
1. The biggest concern associated with the presented manuscript is the sample size for the presented results. The authors work with a transgenic mouse model and have presented results with an experimental sample size ranging between 4-6 which seems very limited and could actually be responsible for the lack of differences observed in some cases (anti-inflammatory cytokine results. Please discuss the caveats associated with the small sample size or the small population of mice used for the experiments.
2. While the use of a small mice population may be justified given the mouse model, it is absolutely unacceptable when the authors continue to use a small sample size when performing experiments with cell lines. There is no clear explanation or description validating the repetition of experiments as the figure legends only describe the sample size. The authors need to clarify if the experiments were indeed repeated and increase the sample size, especially for the experiments performed with the cell lines.
3. Another serious concern associated with the presented manuscript results from the approach the authors adopt in the discussion section. The authors seem to use previous results using other drugs, inhibitors, or molecules to treat ALS and extrapolate their own findings in light of results from these different molecules. While it is important to discuss the current results in light of previous findings the authors tend to take extensive liberty when extending the potential of the observed results in treating ALS or alleviating the symptoms associated with ALS. The authors are advised to temper down the conclusions in light of the observed results based on experiments performed in the current manuscript.
4. The presented model needs to be revisited. The authors did not provide any evidence justifying that MBP application could potentially modulate M2 microglia/macrophage proliferation. Also, Figure 5 only validates the effect of MBP application on LPS-induced proinflammatory cytokines and there is no data presented for anti-inflammatory cytokines. How do the authors classify the results obtained with the BV-2 cells and assign them to M1 vs. M2 type microglia?
5. The authors need to provide a discussion on the potential underlying mechanism that regulates the selectivity of MBP towards inhibition of M1-related pro-inflammatory cytokines in comparison to M2-related anti-inflammatory cytokines. How does MBP selectively inhibit the polarization into M1 microglia/ macrophage in the lumbar spinal cord of mice?
6. The authors should also discuss the caveats of the experimental design in the discussion.
Author Response
"Please see the attachment."

Reviewer 2 Report
In this article, Komugi Tsuruta et al. evaluated MBP's effects on microglial activation in the spinal cords of these mice and in lipopolysaccharide-stimulated BV-2 microglial cells. The findings showed that MBP suppressed the upregulation of M1-microglia/macrophage markers and pro-inflammatory cytokines in the mice, without impacting the increase of M2-microglia/macrophage markers. Additionally, MBP hindered the production of lipopolysaccharide-induced pro-inflammatory cytokines in BV-2 cells. These results underline the therapeutic potential of MBP in ALS model mice, primarily through the selective suppression of M1-related pro-inflammatory cytokines. This article is a continuation of the previous work by the research group and provides valuable references.
Major comments,
1. More M1/M2 polarization markers should be tested, especially Mhcii, CD80/CD83/CD86, NOS2, and ARG1
2. Macrophage proliferation is not a obvious index for macorphage activation. The BV-2 is special because it is a cell line. Please use the primary cells to identify these effects.
Author Response
"Please see the attachment."

Round 2
Reviewer 2 Report
No more comments